# Optimized braces for the treatment of adolescent idiopathic scoliosis: A study protocol of a prospective randomised controlled trial

**Maxence Coulombe**[1,2], **Aymeric Guy**[2,3], **Soraya Barchi**[2], **Hubert Labelle**[2,4]*, **Carl-Éric Aubin**[2,3,4]

1 Department of Medecine, University of Montreal, Montreal, Quebec, Canada, 2 Department of Orthopedics, Sainte-Justine University Hospital Center, Montreal, Quebec, Canada, 3 Department of Mechanical Engineering, Polytechnique Montreal, Montreal, Quebec, Canada, 4 Department of Surgery, University of Montreal, Montreal, Quebec, Canada

* hubert.labelle@umontreal.ca

**Data Availability Statement:** No datasets were generated or analyzed during the current study. All relevant data from this study will be made available upon study completion.

## Abstract

### Introduction

Adolescent Idiopathic Scoliosis (AIS) is a 3D deformity of the spine that affects 3% of the adolescent population. Conservative treatments like bracing aim to halt the progression of the curve to the surgical threshold. Computer-aided design and manufacturing (CAD/CAM) methods for brace design and manufacturing are becoming increasingly used. Linked to CAD/CAM and 3D radiographic reconstruction techniques, we developed a finite element model (FEM) enabling to simulate the brace effectiveness before its fabrication, as well as a semi-automatic design processes. The objective of this randomized controlled trial is to compare and validate such FEM semi-automatic algorithm used to design nighttime Providence-type braces.

### Methods and analysis

Fifty-eight patients with AIS aged between 10 to 16-years and skeletally immature will be recruited. At the delivery stage, all patients will receive both a Providence-type brace optimized by the semi-automatic algorithm leveraging a patient-specific FEM (Test) and a conventional Providence-type brace (Control), both designed using CAD/CAM methods. Biplanar radiographs will be taken for each patient with both braces in a randomized crossover approach to evaluate immediate correction. Patients will then be randomized to keep either the Test or Control brace as prescribed with a renewal if necessary, and will be followed over two years. The primary outcome will be the change in Cobb angle of the main curve after two years. Secondary outcomes will be brace failure rate, quality of life (QoL) and immediate in-brace correction. This is a single-centre study, double-blinded (participant and outcome assessor) randomized controlled trial (RCT).

**Funding:** HL: Canadian Institute of Health Research [H. Labelle et al., 2017-2024, #375116]. URL: https://cihr-irsc.gc.ca/e/193.html CEA: Apogee Canada First Research Excellence Fund [CE Aubin et al., 2016-2026, #CFREF-2015-00008] MC: Transmedtech Institute [Scholarship for Graduate Studies, No grant number associated]. URL: https://transmedtech.org/en/ The funders did not and will not have a role in study design, data collection and analysis, decision to publish, or preparation of the manuscript.

**Competing interests:** MC and AG are cofounders and shareholders of Modulate Technologies Inc. HL is a cofounder and shareholder of Spinologics Inc. CEA has a discovery grant from the Natural Sciences and Engineering Research Council of Canada (NSERC), and a NSERC industrial research grant with Medtronic of Canada paid to the university (outside the scope of the current study). SB declare that the research was conducted in the absence of any commercial or financial relationships that could be construed as a potential conflict of interest. The NSERC grant with Medtronic was not been used to fund this research project. A patent application on the optimization algorithm has been submitted. This does not alter our adherence to PLOS ONE policies on sharing data and materials.

## Trial registration number

ClinicalTrials.gov: NCT05001568.

## Introduction

Adolescent Idiopathic scoliosis (AIS) is a 3D deformity of the spine affecting the alignment of the spine in the coronal, sagittal and transverse planes. Treatments for AIS are usually separated into three main categories: observation, bracing and surgery [1]. Bracing is the most common conservative treatment and is recommended for curves between 25 and 40 degrees in the coronal plane [2]. The aim of bracing is to halt or retard the progression of scoliosis before it progresses to a surgical level (over 45°) [3], by applying external forces to the patient's trunk that will be transmitted to the spine [4].

Multiple types of braces exist with two of the main types being rigid full-time thoracolumbosacral orthoses (TLSO) and nighttime orthoses. TLSO have been extensively studied and a randomised controlled trial by *Weinstein et al.* demonstrated their effectiveness and showed it was proportional to the compliance of the patient [5]. To be effective, a brace should ideally be prescribed to be worn more than 18 hours a day [6]. Studies have shown that TLSO are associated with a lower quality of life (QoL) due to the psychological burden of the brace during the treatment [7, 8]. Recent studies also highlighted the low brace-wear compliance of patients [5, 9]. Alternatives for full-time TLSOs are the less cumbersome nighttime braces, meant to be worn 8 hours/day at night [10]. They feature an overcorrection mechanism to compensate for the lower wear time [11, 12].

Nighttime braces have been shown to prevent curve progression in patients with a main Cobb angle between 15 and 25 degrees [13]. However, they haven't been studied in randomized controlled trials for Cobb angles ranging from 20 to 40 degrees. A recent meta-analysis of case series showed that nighttime braces could be equivalent to TLSO braces [14].

Today braces are mostly designed using topographic scans of the torso and computer-aided design/computer-aided manufacturing (CAD/CAM) [9, 15, 16]. This method can be combined with a patient-specific finite element model (FEM) to simulate different brace designs in a computer environment before manufacturing. This approach was validated by our group in a 2-year RCT of 120 patients with effective results, as the FEM-improved braces provided satisfactory correction while being thinner and using less surface covering area [9]. However, this study was limited by the bias of the orthotists who were influenced over the course of the study by the results of the FEM approach, which reduced the difference in performance between the two cohorts, to the benefit of the treated patients. To reduce this bias and take full advantage of the combined FEM and CAD/CAM approach, we have recently developed an optimization algorithm to design orthotics semi-automatically.

### Objectives

The objective of this trial is to validate a Providence-type brace designed using a patient-specific FEM coupled with a semi-automatic optimization algorithm, and compare its effectiveness to the standard way to design a nighttime Providence-type brace. This will be done by testing the non-inferiority of the immediate and 2-year change in Cobb angle, as well as quality of life as measured by the SRS-22r and the treatment failure rate.

## Methods

### Study design

The study design will be two arms, non-inferiority, double-blinded (patient and outcome assessor) randomized controlled trial. The allocation will be 1:1. The study will be divided in two phases: an immediate effect crossover study and a two-year longitudinal study, as seen in Figs 1 and 2.

In the first phase, each patient will receive both a Control brace (traditional Providence) and a Test brace (optimized FEM Providence). Patients will receive each brace in a randomized order: Control:Test or Test:Control. The immediate in-brace correction will be radiographically assessed and compared.

In the second phase, patients will be randomized to receive either the Control brace or the Test brace. They will wear the same brace type for a two-year period or until skeletal maturity, with follow-up visits every 6 months. The patient will receive a brace renewal if needed, following the assigned group's design protocol.

The study protocol follows the SPIRIT guidelines, and the SPIRIT checklist is included as S1 Checklist.

### Recruitment and informed consent

All patients in this single-centre study will be recruited at an academic pediatric hospital. Patients in the scoliosis clinic who receive a prescription for a Providence brace and meet eligibility criteria will be approached for recruitment. The researchers, who have been trained in how to obtain free and informed consent, will obtain this consent from patients and parents/ legal representatives. After enrollment, patients will be given a unique identifier for anonymization. Recruitment started in September 2021 and it's still ongoing.

### Eligibility

**Inclusion criteria.**

a. Diagnosis of AIS.

b. Age between 10 and 16 years.

c. Risser between 0 and 2.

d. Cobb angle of the main curve between 20˚ and 40˚.

e. Premenarchal or no more than 18-year postmenarchal for female patients [17].

**Exclusion criteria.**

a. History of cardiovascular disease

b. History of neuropathic disease

c. History of musculoskeletal disease of the inferior limb

d. Spondylolisthesis

e. Pregnancy

| | Study Period | | | | |
|---|---|---|---|---|---|
| | **Enrolment** | **Allocation** | **Post-allocation** | | **Close-out** |
| **TIMEPOINT\*\*** | *Initial visit (t₁)* | **Brace delivery (t₁+1 month)** | **Follow-up visit (every 6 month after t₁)** | **Renewal brace delivery visit if necessary (1 month after follow-up visit)** | *End of study (t₁+2 years)* |
| **ENROLMENT:** | | | | | |
| **Eligibility screen** | X | | | | |
| **Informed consent** | X | | | | |
| *Out-of-brace radiographs* | X | | | | |
| *Torso scan* | X | | | | |
| **Allocation** | | X | | | |
| **INTERVENTIONS:** | | | | | |
| *Control* | | X | ◆————————————————◆ | | |
| *Test* | | X | ◆————————————————◆ | | |
| **ASSESSMENTS:** | | | | | |
| *Out-of-brace Cobb angle* | X | | X | | X |
| *Immediate in-brace Cobb angle* | | X | | X | |
| *QoL SRS-22r* | X | | X | | X |
| **Treatment compliance** | | | X | | X |
| *Treatment failure* | | | X | | X |

**Fig 1. SPIRIT schedule of enrolment, interventions, and assessments.**

## Randomization

Patients will be randomized twice: Patients will first be randomized to determine the sequence of the brace they will try. Allocation for the first randomization will be 1:1 for either the

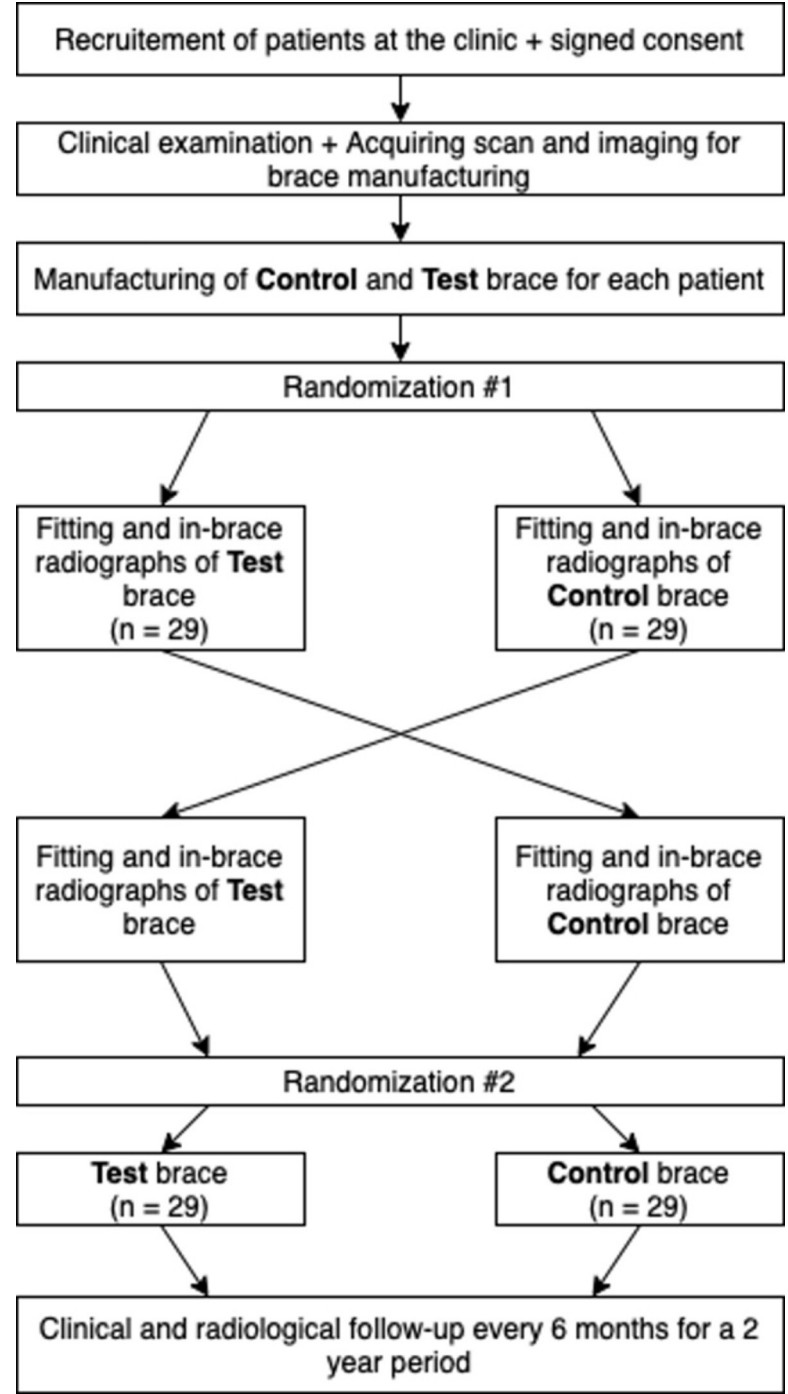

**Fig 2. Study design.**

sequence Test:Control or the sequence Control:Test. The second randomization will be to determine the brace that will be assigned to pursue the long-term treatment. The allocation will be 1:1 for either the Test brace or the Control brace. The randomization will be done by blocks of 2, 4 or 6. The individual randomization sequence will be concealed in separate sealed

envelopes independently from the research team. The treatment assignement will be done at the brace delivery (visit 2) by the research team.

For ethical reasons, if one of the braces during the crossover phase has an in-brace correction of the primary curve of more than 5 degrees compared to the other brace, the participant will be switched to the brace with the better in-brace correction.

## Blinding

The treating physician, patients and outcome assessors will be blinded to the randomization status of the participants. The physician and patient will not be able to identify the braces and thus the group assignation. The outcome assessors will be independent from the research team. The authors of the protocol have access to participants data.

## Interventions

Two brace designs will be created in this trial: a control Providence brace (Control) and an optimized FEM Providence-type brace (Test).

**Providence brace–control.** The control Providence type brace will be designed by an orthotist with more than 20 years of experience. The orthotist will scan the patient torso in the supine position using the MSoft software (TechMed3D, Lévis, Canada). The scan will be uploaded in a CAD software (Rodin4D, Merignac, France). The orthotist will then design the Providence type brace by sculpting the shape of the brace and by adding areas of pressure and reliefs in the software. The final model will be exported to a numerical milling machine (Model C, Rodin4D) that will carve a polyurethane foam block to use for thermoforming. Upon delivery, the orthotist will make the final adjustments on the brace to ensure a comfortable fit.

**Optimized FEM providence brace–test.** The test Providence brace will be designed by the researchers and the orthopedist. As described by Cobetto et al. [18], the patient's biplane radiographs will be used to build a 3D reconstruction of the spine from C7 to the coccyx, as well as the pelvis and the rib cage. A scan of the patient torso will be taken in the standing position using MSoft software (TechMed3D, Lévis, Canada). The 3D reconstruction and the scan will then be superimposed and imported into ANSYS 20.1 software (ANSYS, Canonsburg, USA) to create a patient-specific FEM. The intervertebral discs and osseous structures such as thoracic and lumbar vertebrae, ribs, sternum, and pelvis will be represented by 3D elastic beam elements. Ligaments, soft tissues, and joints will be modelled by tension-only springs. External soft tissues and their interaction with the brace will be represented by shells and surface-to-surface contact elements respectively. Mechanical properties of all anatomical structures are taken from cadaveric studies, and an estimate of patient flexibility will be factored in. A previously described optimization process computes the gravitational and stabilizing muscular forces to obtain a loaded geometry corresponding to the standing radiographs [19].

An initial brace design model is automatically generated from the patient's skin geometry and is used as input of a surrogate optimization algorithm. During optimization, a new brace topology is generated at every iteration and imported into the patient-specific FEM. A supine position on a mattress is simulated [20] with brace donning and tightening. Based on these results, patient growth is then simulated for a period of two years, with the vertebral growth rate modulated in response to the simulated stresses acting on the vertebral growth plates [21]. 3D correction metrics (Cobb angle MT & TL/L, TK/LL, mean axial rotation) are extracted from the corrected spine resulting from the two simulations (immediate in-brace supine and 2-year out-of-brace post-growth) and combined as a weighted sum to create the optimization's objective function. The optimization process runs for hundreds of iterations to converge

towards the optimal brace topology maximizing immediate and 2-year correction. The patient's simulated corrected spine will be compared to the clinical radiographs to further validate the model for this context of use specifically.

The final design will be revised by an orthopedist. Both braces will be manufactured by the same method. Each brace (Control or Test) will be equipped with a thermal compliance monitoring device (iButton, Maxim Integrated Products Inc, Sunnyvale, CA). In each instance, the patient will be instructed to wear the brace at night when sleeping and compliance will be followed-up every 6 months.

**Assessment and management.** Low-dose biplanar (coronal and sagittal) radiographs (EOS system, EOS imaging, Paris) will be acquired out-of-brace at the first visit, as well as at the follow-up visits every six months prior to seeing the orthopedic surgeon. At the initial and follow-up visits, the orthopedic surgeon will do a physical examination of the back and assess their general condition. An anterior-posterior (AP) x-ray will also be taken each time a new brace is delivered, to verify the fit. At each visit including the first one, the patient will be asked to fill the SRS-22r questionnaire. Other conservative treatments are not prohibited for this trial.

## Outcome measurements

**Primary outcome.** The primary outcome of this study will be the changes in Cobb angle out-of-brace. The measurements will be made at the first visit (baseline), and at the one-year and two-year follow-ups, with an in-house software (Clindexia) from the 3D reconstruction.

**Secondary outcome.** The study will have three secondary outcomes:

- Treatment failure rate: Treatment failure rate is defined as having a Cobb angle progress by more than 5° compared to the baseline measurement or exceeding a value of 45°. The results will be compared between the groups.

- Quality of Life (QoL)–SRS-22r: The quality of life will be measured using the Scoliosis Research Society (SRS) outcome assessment tool: SRS-22r. The French and English versions of the questionnaire will be used depending on the language spoken by the participant. The results will be compared between groups.

- Immediate In-Brace Cobb Angle: Each patient will try both a control and a test brace, and an in-brace radiograph will be taken. The Cobb angle measurement will be made directly on the radiographic images. The immediate in-brace correction in Cobb angle will be compared between groups.

## Data management

Each patient will receive a unique research identifier and all the data associated with the patient will have this identifier. All data, including the patient history, type of brace, radiographs, 3D files and questionnaires will be stored in a Microsoft Access relational database (Microsoft, Redmond, USA). This database will be hosted on a secure server. Data from participants who discontinue or deviate from intervention protocols will be collected from the patient medical records. The study is done in compliance with a quality management system for medical devices (ISO 13485:2016). Data analysis and monitoring will be done by the Applied Clinical Research department of the academic hospital, this department is independent from the sponsor and competing interests.

### Safety monitoring and adverse events

All participants will receive a brace that will be approved by a healthcare professional, either the orthotist for the control brace or the orthopaedist for the test brace. The study has strict failure criteria to protect participants. If a participant meets a failure criterion, he/she will receive other treatment options. Any unexpected or adverse events will also be monitored and reported. All patients will have the coordinates of the research team and the orthopaedic clinic. They will also be able to make an appointment sooner than the 6-month follow-up in case of a problem. The trial will be externally audited by the Ethics and Research committee of the hospital.

### Sample size calculation

The sample size was calculated to answer our primary hypothesis. The sample size was calculated with a significance level of 5%, a power of 80%, an allocation ratio of 1 and a non-inferiority margin of 5˚. With an estimated 15% of participants loss, the sample size calculated was 29 participants per group for a total of 58. Automatic scheduling of follow-up visits and reminders will be used to reduce participants loss.

### Statics analysis

There will be both an intention to treat (ITT) and intention of protocol (ITP) analysis. The ITT analysis will include all patients that have been randomized. Patients generally wear their nighttime braces 90.7% of the 8 hours generally prescribe[22], for this reason the ITP analysis will include patient that wore their brace an average of 7.2 hours per day. This analysis will determine the effect of the brace if it is worn with a good compliance, allowing to target future improvements. For both analysis, the continuous data will be compared between the test and control group with a non-inferior one-sided two-sample t-test, this includes the out-of-brace Cobb angle, the immediate in-brace Cobb angle and the SRS-22r scores. A survival analysis using the log-rank test will be done to analyze the failure rate will be presented on a Kaplan-Meier curve. In case of missing data, a worst-case single imputation will be used. Point estimates and 95% confidence interval will be provided for each result. The statistical analysis will be performed using the R programming language.

### Ethics and dissemination

This trial (2022–3437) has been approved by the Comité d'éthique de la recherche du CHU Sainte-Justine (CHU Sainte-Justine Research Ethics Committee) on July 8, 2021 and registered on ClinicalTrials.gov (NCT05001568). Written informed consent will be obtained for all participants in the study. Results of the trial will be submitted for publication in a peer-reviewed journal and as conference presentations. Changes to the protocol will be communicated with the research and ethics committee, and also made available on ClinicalTrials.gov.

### Status and timeline of the study

The study is currently recruiting participants. The estimated completion date for the study is January 2025.

## Discussion

This article presented the need, as well as the design, of a prospective randomized controlled trial. This methodology will allow us to validate the use of an optimization algorithm that iterates based on a previously validated FEM [9, 18, 23]. This algorithm has the potential of

improving clinical outcomes, as well as reducing the time it takes for the orthotist to design nighttime braces.

The outcomes evaluated will be clinically relevant, they include the immediate in-brace Cobb angle correction as well as the long-term out-of-brace Cobb. It will also include QoL questionnaires that will assess both the physical and mental health of the participants.

The patients will be recruited based on the standardized SRS bracing criteria with primary curve angle modified to account for angle measurement variability [24]. This study also presents a new methodology for RCTs on braces, by comparing the immediate correction of multiple braces on the same patients, as well as the long-term correction. This is the first methodology, to our knowledge, that allows the direct comparison of two brace models using the immediate in-brace correction as a surrogate for the primary outcome of this study [25], while also having a two-year follow-up. Patients will be treated by an integrated team comprising of orthopedic surgeons, orthotists, nurses, and physiotherapists.

## Supporting information

**S1 Checklist. SPIRIT checklist.**
(DOCX)

**S1 Protocol. Protocol of the study v1.1 [January 2023] (translated from french to english).**
(PDF)

**S1 Document. Informed consent form (translated from french to english).**
(PDF)

## Author Contributions

**Conceptualization:** Maxence Coulombe, Aymeric Guy, Soraya Barchi, Hubert Labelle, Carl-Éric Aubin.

**Funding acquisition:** Hubert Labelle, Carl-Éric Aubin.

**Methodology:** Maxence Coulombe, Aymeric Guy, Soraya Barchi, Hubert Labelle, Carl-Éric Aubin.

**Project administration:** Maxence Coulombe, Soraya Barchi.

**Supervision:** Hubert Labelle, Carl-Éric Aubin.

**Writing – original draft:** Maxence Coulombe.

**Writing – review & editing:** Aymeric Guy, Soraya Barchi, Hubert Labelle, Carl-Éric Aubin.

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
