## [Decision Letter · Decision Letter 0]

1 Aug 2023

PONE-D-23-11353Optimized braces for the treatment of adolescent idiopathic scoliosis: a study protocol of a prospective randomised controlled trialPLOS ONE

Dear Dr. Hubert,

Thank you for submitting your manuscript to PLOS ONE. After careful consideration, we feel that it has merit but does not fully meet PLOS ONE’s publication criteria as it currently stands. Therefore, we invite you to submit a revised version of the manuscript that addresses the points raised during the review process.

We look forward to receiving your revised manuscript.

Kind regards,

Filippo Migliorini MD, PhD, MBA

Academic Editor

PLOS ONE

Journal Requirements:

I have read the journal's policy and the authors of this manuscript have the following competing interests: MC and AG are cofounders and shareholders of Modulate Technologies Inc. HL is a cofounder and shareholder of Spinologics Inc. CEA has a discovery grant from the Natural Sciences and Engineering Research Council of Canada, and a R&D grants from Medtronic paid to the university. SB declare that the research was conducted in the absence of any commercial or financial relationships that could be construed as a potential conflict of interest.  

We note that you received funding from a commercial source.

4. We note that the original protocol that you have uploaded as a Supporting Information file contains an institutional logo. As this logo is likely copyrighted, we ask that you please remove it from this file and upload an updated version upon resubmission.

Additional Editor Comments:

please follow the SPIRIT guidelines: re-adapt the whole manuscript, state in a sentence that you follow those guidelines and add its related reference, and add the checklist filled in each part as supplementary material. Thank you. Regards, Filippo Migliorini

Reviewers' comments:

Reviewer's Responses to Questions

**Comments to the Author**

1. Does the manuscript provide a valid rationale for the proposed study, with clearly identified and justified research questions?

Reviewer #1: Yes

Reviewer #2: Yes

Reviewer #3: Yes

2. Is the protocol technically sound and planned in a manner that will lead to a meaningful outcome and allow testing the stated hypotheses?

Reviewer #1: Yes

Reviewer #2: Yes

Reviewer #3: Yes

3. Is the methodology feasible and described in sufficient detail to allow the work to be replicable?

Reviewer #1: Yes

Reviewer #2: Yes

Reviewer #3: Yes

4. Have the authors described where all data underlying the findings will be made available when the study is complete?

Reviewer #1: No

Reviewer #2: Yes

Reviewer #3: Yes

5. Is the manuscript presented in an intelligible fashion and written in standard English?

Reviewer #1: Yes

Reviewer #2: Yes

Reviewer #3: Yes

6. Review Comments to the Author

You may also provide optional suggestions and comments to authors that they might find helpful in planning their study.

Reviewer #1: In this study protocol, a two-arm non-inferiority randomized control trial is being proposed to compare and validate a FEM semi-automatic algorithm used in designing back braces. The target accrual is 58. All patients will receive both the experimental and control braces and undergo immediate radiographic evaluations. Patients will then be randomized to keep either the experimental or control brace. The primary outcome is the change in Cobb angle of the main curve at two years. Secondary outcomes will be brace failure, quality of life, and immediate in-brace correction.

Minor revisions:

1- Line 224: Identify the software that will be used to capture the data.

2- Provide a statistical analysis plan for the quality of life and immediate brace correction outcomes.

3- Line 250 states that the failure rate will be illustrated on a Kaplan-Meier curve. Indicate if the two groups will be compared using inferential statistics. If so, state the testing method that will be used to make this comparison. Will point estimates and 95% confidence intervals be provided?

Reviewer #2: Thank you very much for the opportunity to review the submitted article "Optimized braces for the treatment of adolescent idiopathic scoliosis: a study protocol of a prospective randomised controlled trial".

The planned study is evaluating a clinically and scientifically important topic prospectively.

One concern is related to the low compliance reported for braces for the treatment of scoliosis and possible drop-out. How do the authors plan to avoid this problem or plan to deal with it if it occurs (e.g longer recruitment interval, multi-centric approach)?

First, as the study focuses on pediatric/adolescent patients, informed consent must be obtained from participants AND legal representatives. The authors planned to obtain informed consent from patients "and/or parents". Please modify.

Furthermore, the authors submitted informations that a conflict of interest is present in some individuals involved in the study. The authors should provide more detailed data on the process of blinding and to assure that individuals with a conflict of interest are not involved in the measurement of clinical outcomes. Please comment.

Reviewer #3: Dear author,

first of all, I congratulate you on the work you have done. This is a very interesting study regarding improving of braces in the treatment of adolescent idiopatic scoliosis.

I think the quality of your paper could be improved in accordance with the following suggestions:

Introduction:

- Line 75: "To be effective, a brace should ideally be worn 20 to 23 hours a day."

Could you please cite a reference/study that support it?

- Line 76: "Studies have shown that TLSO are associated with a lower quality of life during the treatment"

Could you please explain it? You might find some supporting literature here:

PMID 33098493

Inclusion criteria:

- why do you exclude premenarchal or no more than 18-year postmenarchal for female patients? There is a particular reason?

- Could BMI - or to be more exact heigt and weight - be or not be inclusion criteria of the study? The same question could also regard the familiarity for AIS.

7. PLOS authors have the option to publish the peer review history of their article (what does this mean?). If published, this will include your full peer review and any attached files.

Reviewer #1: No

Reviewer #2: **Yes: **Priv.-Doz. Dr. med. Christian Weber

Reviewer #3: No

---

## [Author Response · Author response to Decision Letter 0]

6 Sep 2023

Response to Reviewers

Manuscript PONE-D-23-11353 Optimized braces for the treatment of adolescent idiopathic scoliosis: a study protocol of a prospective randomised controlled trial.

The authors sincerely appreciate the time and effort dedicated to reviewing our manuscript. The manuscript has been revised to address the editor and reviewers’ comments and concerns. Reviewers’ comments are in italics followed by detailed responses in indented paragraphs. All changes referred to below have been made to the revised manuscript and highlighted for easy identification.

Responses to Editors Comments:

and https://journals.plos.org/plosone/s/file?id=ba62/PLOSOne_formatting_sample_title_authors_affiliations.pdf

The manuscript was updated to follow PLOS One style requirements. 

This includes the formatting of affiliations, the capitalization of tittles and the names of figures and files. References have been changed from parenthesis to brackets. Supporting information section has been moved at end of document and reformatted to matched the PLOS One template (https://journals.plos.org/plosone/s/supporting-information).

I have read the journal's policy and the authors of this manuscript have the following competing interests: MC and AG are cofounders and shareholders of Modulate Technologies Inc. HL is a cofounder and shareholder of Spinologics Inc. CEA has a discovery grant from the Natural Sciences and Engineering Research Council of Canada, and a R&D grants from Medtronic paid to the university. SB declare that the research was conducted in the absence of any commercial or financial relationships that could be construed as a potential conflict of interest. 

We note that you received funding from a commercial source.

The competing statement has been amended to this: I have read the journal's policy and the authors of this manuscript have the following competing interests: MC and AG are cofounders and shareholders of Modulate Technologies Inc. HL is a cofounder and shareholder of Spinologics Inc. CEA has a discovery grant from the Natural Sciences and Engineering Research Council of Canada (NSERC), and a NSERC industrial research grant with Medtronic of Canada paid to the university (outside the scope of the current study). SB declares that the research was conducted in the absence of any commercial or financial relationships that could be construed as a potential conflict of interest. The NSERC grant with Medtronic was not used to fund this research project. A patent application on the optimization algorithm has been submitted. This does not alter our adherence to PLOS ONE policies on sharing data and materials. [Line 34]

The amended competing statement has been added to the manuscript. It states that the NSERC industrial research grant with Medtronic of Canada was not used for this project. It declares a patent application on the optimization algorithm. It also reiterates that this doesn’t alter our adherence to PLOS One policies on sharing. 

The Ethics and dissemination section has been removed from the revised version of the abstract. It is now only mentioned in the Methods section of the manuscript. [Line 295]

4. We note that the original protocol that you have uploaded as a Supporting Information file contains an institutional logo. As this logo is likely copyrighted, we ask that you please remove it from this file and upload an updated version upon resubmission.

The institutional logo from the original protocol have been removed. [Line 422]

The reference list has been reviewed and no retracted paper was found in Pubmed. [Line 336]

6. Please follow the SPIRIT guidelines: re-adapt the whole manuscript, state in a sentence that you follow those guidelines and add its related reference, and add the checklist filled in each part as supplementary material.

The Spirit checklist was revised, and missing elements were added to the manuscript and referenced in the Spirit Checklist [Line 421]. A sentence was added to state that we follow Spirit guidelines [Line 132].

Responses to Comments to Authors

Point #4: Have the authors described where all data underlying the findings will be made available when the study is complete?

In the revised version, we have specified that the anonymize data points will be made available in a public repository for academic purposes. [Line 44]

Responses to Reviewers’ Comments

Reviewer #1 (minor revision):

1- Line 224: Identify the software that will be used to capture the data.

The software used to capture the data was added in the revised manuscript. [Line 259]: “All data, including the patient history, type of brace, radiographs, 3D files and questionnaires will be stored in a Microsoft Access relational database (Microsoft, Redmond, USA).”

2- Provide a statistical analysis plan for the quality of life and immediate brace correction outcomes

Details about the statistical analysis plan for the quality of life (SRS-22r scores) and the immediate in-brace correction outcomes were added the Statistical analysis section. [Line 288]: “ For both analysis, the continuous data will be compared between the test and control group with a non-inferior one-sided two-sample t-test, this includes the out-of-brace Cobb angle, the immediate in-brace Cobb angle and the SRS-22r scores. A survival analysis using the log-rank test will be done to analyze the failure rate will be presented on a Kaplan-Meier curve. In case of missing data, a worst-case single imputation will be used. Point estimates and 95% confidence interval will be provided for each result. The statistical analysis will be performed using the R programming language.”

3- Line 250 states that the failure rate will be illustrated on a Kaplan-Meier curve. Indicate if the two groups will be compared using inferential statistics. If so, state the testing method that will be used to make this comparison. Will point estimates and 95% confidence intervals be provided?

The statistical plan for the survival analysis (Kaplan-Meier curve) was added. [Line 290] We also confirmed that point estimates and confidence intervals will be provided. [Line 292]

Reviewer #2:

1- One concern is related to the low compliance reported for braces for the treatment of scoliosis and possible drop-out. How do the authors plan to avoid this problem or plan to deal with it if it occurs (e.g longer recruitment interval, multi-centric approach)?

The dropout rate at our center is low and measures are in place to reduce this occurrence, such as automatic scheduling of follow-up visits and reminders. As for the low compliance, this will be considered part of the treatment, and evaluated in the per protocol analysis with the compliance monitor data. Sentences were added in the Revised manuscript [Lines 280-281] to detail how we plan to deal with this issue.

2- First, as the study focuses on pediatric/adolescent patients, informed consent must be obtained from participants AND legal representatives. The authors planned to obtain informed consent from patients "and/or parents". Please modify.

As suggested, the following sentence was added in the Revised manuscript: “The researchers, who have been trained in how to obtain free and informed consent, will obtain this consent from patients and parents/legal representatives.” [Line 140]

3- Furthermore, the authors submitted informations that a conflict of interest is present in some individuals involved in the study. The authors should provide more detailed data on the process of blinding and to assure that individuals with a conflict of interest are not involved in the measurement of clinical outcomes. Please comment.

In the Revised manuscript, the following details were added about the blinding process. We also clarified that the outcome assessors will be independent from the research team: “The physician and patient will not be able to identify the braces and thus the group assignation. The outcome assessors will be independent from the research team.” [Line 173]

Reviewer #3:

Dear author, first of all, I congratulate you on the work you have done. This is a very interesting study regarding improving of braces in the treatment of adolescent idiopatic scoliosis. I think the quality of your paper could be improved in accordance with the following suggestions…

We thank this reviewer for the positive appreciation of our study. 

1- Introduction: - Line 75: "To be effective, a brace should ideally be worn 20 to 23 hours a day." Could you please cite a reference/study that support it?

As suggested, a reference was added in the Revised introduction section, the following sentence was modified to: “To be effective, a brace should ideally be prescribed to be worn more than 18 hours a day, based on the BrAIST Trial recommendations (Weinstein SL, Dolan LA, Wright JG, Dobbs MB. Design of the Bracing in Adolescent Idiopathic Scoliosis Trial (BrAIST): Spine. 2013 Oct;38(21):1832–41.)”. [Line 84]

2- Introduction - Line 76: "Studies have shown that TLSO are associated with a lower quality of life during the treatment". Could you please explain it? You might find some supporting literature here: PMID 33098493

In the Revised manuscript, we modified the following sentence and a reference to explain that the lower quality of life was due to the psychological burden of the brace treatment: “Studies have shown that TLSO are associated with a lower quality of life (QoL) due to the psychological burden of the brace during the treatment (Cheung PWH, Wong CKH, Cheung JPY. An Insight Into the Health-Related Quality of Life of Adolescent Idiopathic Scoliosis Patients Who Are Braced, Observed, and Previously Braced. Spine. 2019 May 15;44(10):E596–605.; 

Wang H, Tetteroo D, Arts JJC, Markopoulos P, Ito K. Quality of life of adolescent idiopathic scoliosis patients under brace treatment: a brief communication of literature review. Qual Life Res Int J Qual Life Asp Treat Care Rehabil. 2021 Mar;30(3):703–11.)”. [Line 85]

3- Inclusion criteria:

- why do you exclude premenarchal or no more than 18-year postmenarchal for female patients? There is a particular reason?

Based on Little et al 2000 (Little DG, Song KM, Katz D, et al. Relationship of peak height velocity to other maturity indicators in idiopathic scoliosis in girls. J Bone Joint Surg Am 2000;82-A:685–93.), the exclusion criteria of more than 18-month postmenarchal was chosen as an indication that skeletal maturity was reached and that the brace treatment would not be appropriate in that group of patients. Premenarchal patient and patient who had their menarche less than 18-month prior are included in the study. In the revised manuscript, the reference to include this relevant point raised by this reviewer [Line 150] 

- Could BMI - or to be more exact heigt and weight - be or not be inclusion criteria of the study? The same question could also regard the familiarity for AIS.

BMI, weight and height were not chosen as inclusion/exclusion criteria in order to maximize the external validity of the study. In the publication resulting from the study, we will present the available data on BMI. We also didn’t use a family history of AIS as inclusion/exclusion criteria in order to maximize the external validity of the study.

---

## [Editor Report · Decision Letter 1]

12 Sep 2023

Optimized braces for the treatment of adolescent idiopathic scoliosis: a study protocol of a prospective randomised controlled trial

PONE-D-23-11353R1

Dear Dr. Hubert,

We’re pleased to inform you that your manuscript has been judged scientifically suitable for publication and will be formally accepted for publication once it meets all outstanding technical requirements.

Kind regards,

Filippo Migliorini MD, PhD, MBA

Academic Editor

PLOS ONE

Additional Editor Comments (optional):

Well done! Regards, Filippo Migliorini
---

## [Editor Report · Acceptance letter]

3 Oct 2023

PONE-D-23-11353R1 

Optimized braces for the treatment of adolescent idiopathic scoliosis: a study protocol of a prospective randomised controlled trial 

Dear Dr. Labelle:

I'm pleased to inform you that your manuscript has been deemed suitable for publication in PLOS ONE. Congratulations! Your manuscript is now with our production department. 

Kind regards, 

on behalf of

Dr Filippo Migliorini 

Academic Editor

PLOS ONE